# MTMC: Generalized Category Discovery via Maximum Token Manifold Capacity

## Abstract

Identifying previously unseen data is crucial for enhancing the robustness of deep learning models in the open world. Generalized category discovery (GCD) is a representative problem that requires clustering unlabeled data that includes known and novel categories. Current GCD methods mostly focus on minimizing intra-cluster variations, often at the cost of manifold capacity, thus limiting the richness of within-class representations. In this paper, we introduce a novel GCD approach that emphasizes maximizing the token manifold capacity (MTMC) within class tokens, thereby preserving the diversity and complexity of the data's intrinsic structure. Specifically, MTMC's efficacy is fundamentally rooted in its ability to leverage the nuclear norm of the singular values as a quantitative measure of the manifold capacity. MTMC enforces a richer and more informative representation within the manifolds of different patches constituting the same sample. MTMC ensures that, for each cluster, the representations of different patches of the same sample are compact and lie in a low-dimensional space, thereby enhancing discriminability. By doing so, the model could capture each class's nuanced semantic details and prevent the loss of critical information during the clustering process. MTMC promotes a comprehensive, non-collapsed representation that improves inter-class separability without adding excessive complexity.

## 1 Introduction

Machine learning models encounter substantial challenges when deployed in real-world settings due to the intractability of objects in the open world (Zhou et al., 2022; Sarker, 2021; Weiss et al., 2016). The diversity of real-world objects exceeds the scope of data collected for training (Wu et al., 2024), and labeled data covers even fewer categories. Traditional deep learning models, trained on predefined categories, are ill-equipped to handle new category samples. To enhance the reliability of model deployment in real-world scenarios, open-world learning has emerged, aiming to identify and categorize unknown samplese (Han et al., 2019; Geng et al., 2020; Vaze et al., 2022) in new environments.

A plethora of approaches have been proposed to identify and categorize unknown samples, such as open-set recognition (OSR) (Geng et al., 2020) and novel class discovery (NCD) (Han et al., 2019). However, OSR treats all unknown samples as a single category. On the other hand, NCD relies on a strong assumption that all unlabeled samples encountered come from new classes. To relax this assumption, Generalized Category Discovery (GCD) (Vaze et al., 2022) permits the presence of known classes within unlabeled data. GCD relies on contrastive learning (Choi et al., 2024) or prototype learning (Wen et al., 2023) to reduce the distance between semantically identical samples in the embedding space. However, current approaches face a significant challenge, *i.e.,* the compressed inter-class distribution may lead to the loss of useful information. This results in each cluster being unable to fully represent the semantic details within a class, leading to bias within the feature space, which is detrimental to category discovery. The reason is that bias prevents the inter-class decision boundaries from aligning with the boundaries between real-world categories, making it impossible for the model to accurately separate clusters during the discovery of categories (Figure 5 demonstrates that incomplete intra-class representations result in low clustering accuracy).

To this premise, we challenge the status quo by raising an open question: *Can deep models accurately separate new semantics during the category discovery by enhancing the completeness of intra-class representations?*

The GCD aims to partition data points into distinct clusters, which are distributed on low-dimensional manifolds (Souvenir & Pless, 2005; Wah et al., 2011) within high-dimensional spaces. Recently, Maximum Manifold Capacity Representations (MMCR) (Yerxa et al., 2023; Schaeffer et al., 2024; Isik et al., 2023) have sought to learn representations by examining the separability of manifolds. In this context, manifolds containing views of the same scene are both compact and low-dimensional, while manifolds corresponding to different scenes are maximally separated. Building on this concept, we introduce Maximum Class Token Manifold Capacity (MTMC). Specifically, we associate low intra-class representation completeness with low manifold capacity. Our research narrows the focus from the entire feature space to the intra-class feature space, examining manifold capacity at a more granular token level. We consider the representation of a sample as its manifold, with the sample representation in GCD derived from the class token provided by Vision Transformer (ViT) (Dosovitskiy, 2020). Under the attention mechanism, the class token refines the patch features, thus serving as a proxy for the sample manifold. Given that a comprehensive and information-rich class token manifold necessitates a large capacity, we measure manifold capacity using the nuclear norm of the class token and aim to maximize this norm. MTMC enhances the completeness of sample representation, enabling clusters to capture more intra-class semantic details while preventing dimensionality collapse, thus improving inter-class separability accuracy.

Our contributions can be summarized as follows:

- We propose a method called MTMC to enhance representation completeness, thereby empowering the model for generalized category discovery. We theoretically analyze the effectiveness of MTMC as a means to address dimensionality collapse and enhance representation quality.

- We increase the capacity of the class token manifold by maximizing the nuclear norm of the singular value kernel of the class token, allowing clusters to represent more intra-class semantic details.

- MTMC is simple to implement. Experiments on coarse-grained and fine-grained datasets prove the effectiveness of precision in category discovery and accuracy in estimating the number of categories.

## 2 PRELIMINARY AND MOTIVATION

### 2.1 NOTATION OF GCD

For each dataset, consider a labeled subset $\mathcal{D}_l = \{(\mathbf{x}_i^l, y_i^l)\} \subset \mathcal{X} \times \mathcal{Y}_l$ and an unlabeled subset $\mathcal{D}_u = \{(\mathbf{x}_i^u, y_i^u)\} \subset \mathcal{X} \times \mathcal{Y}_u$. Only known classes can be found in $\mathcal{D}_l$, while $\mathcal{D}_u$ encompasses known and novel classes, translating to $\mathcal{Y}_l = \mathcal{C}_{known}$ and $\mathcal{Y}_u = \mathcal{C}_{known} \cup \mathcal{C}_{novel}$. The task of models involves clustering on both the known and novel classes in $\mathcal{D}_u$. The number of novel classes represented as $K_{novel}$ can be determined beforehand (Vaze et al., 2022; Pu et al., 2023; Zhao et al., 2023). The functions $f(\cdot)$ and $g(\cdot)$ perform as the feature extractor and projection head, respectively. Both the feature $\mathbf{h}_i = f(\mathbf{x}_i)$ and the projected embedding $\mathbf{z}_i = g(\mathbf{h}_i)$ are under L-2 normalization.

### 2.2 OPTIMIZATION OBJECTIVE OF GCD

For compact clustering, GCD has two universally applicable components (Appendix A.xx), formally represented as supervised and unsupervised contrastive learning $\mathcal{L}_{\text{sup}} + \mathcal{L}_{\text{unsup}}$, and prototype learning $\mathcal{L}_{\text{proto}}$. The goal is to pull similar samples closer in feature space strongly. Their optimization objectives are summarized as follows: the pioneering work GCD (Vaze et al., 2022) minimizes $\mathcal{L}_{\text{GCD}} = \mathcal{L}_{\text{sup}} + \mathcal{L}_{\text{unsup}}$, which conducts contrastive learning on samples within a mini-batch, and performs semi-supervised clustering after training. $\mathcal{L}_{\text{CMS}} = \mathcal{L}_{\text{sup}} + \mathcal{L}_{\text{unsup}} + \mathcal{L}_{\text{proto}}$, CMS (Choi et al., 2024) incorporates mean-shift, implicitly introducing a prototype by including the feature mean of samples into contrastive learning. SimGCD (Wen et al., 2023) constructs a prototype classifier and performs semi-supervised learning like FixMatch and self-distillation with $\mathcal{L}_{\text{SimGCD}} = \mathcal{L}_{\text{proto}}$.

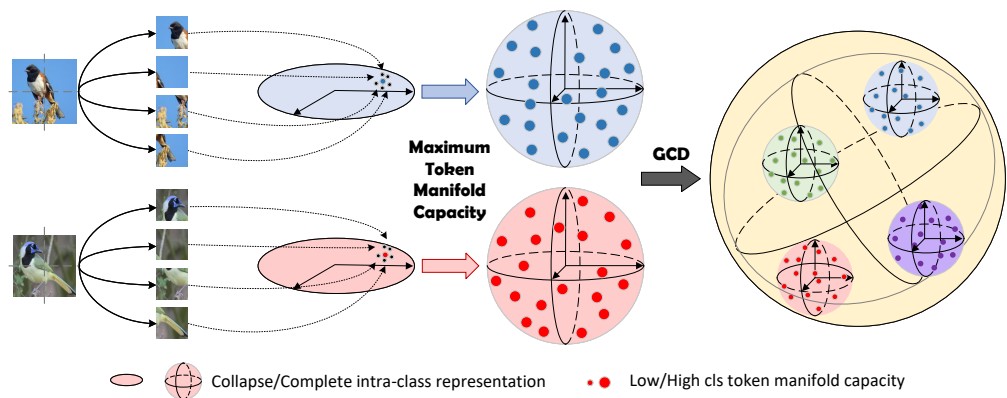

Figure 1: Overview of Maximum Token Manifold Capacity.

The schemes above yield good clustering results, but they overly focus on forming individual class clusters, neglecting the incomplete intra-class representation, which is insufficient to cover the real distribution and represents a low manifold capacity.

### 2.3 MOTIVATION

Manifold capacity can be regarded as a sample-level distribution range. For low-dimensional data, manifold capacity can be intuitively understood as a combination of manifold radius and dimension (Yerxa et al., 2023). Maximum Manifold Capacity Representation can achieve self-supervised learning by maximizing the capacity of the manifold of samples and their augmented views, causing samples to uniformly fill the feature space and similar samples to be closer.

Before formally introducing the details of the methodology, we briefly discuss the motivation: (1) Since GCD already has optimization objectives like $\mathcal{L}_{\text{SimGCD}}$, $\mathcal{L}_{\text{CMS}}$ that bring the embedding distances of similar samples closer, overly compact clusters represent an incomplete representation. Therefore, we aspire to enhance the feature completeness within the intra-class, ensuring its range is sufficient to cover the real distribution, to promote more accurate clustering, as correct and rich clusters help shape more reasonable inter-class decision boundaries. (2) Inspired by the sample-level MMCR, maximizing the manifold capacity of samples and their augmented views can separate samples. Since our research point is the richness of intra-class representation, maximizing the manifold capacity at the token level after cutting samples into patches would increase the embedding distance between different semantic patches within a cluster, enhancing the intra-class manifold capacity. (3) Estimating token-level manifold capacity is key, we trace the formation of token embeddings for various attributes and determine that maximizing the class token manifold capacity can reasonably and succinctly enhance the completeness of intra-class representation.

## 3 METHODOLOGY

As shown in Figure 1, Maximum Token Manifold Capacity is pithy. For simplicity, we use `[cls]` to represent the class token and `[vis]` to represent visual/patch tokens. In Subsection 3.1, we trace the formation process of `[cls]` and `[vis]`, and identify `[cls]` as the sample centroid, also providing the definition of class token manifold extent, which is strongly correlated with capacity. In Subsection 3.2, we introduce the optimization objective of maximum class token manifold capacity and offer a concise code illustration.

### 3.1 EXTENT OF CLASS TOKEN MANIFOLD

We introduce the concept of "sample centroid" without imposing restrictions on network structures, whether they are CNNs or Transformers. In the GCD task, the backbone network is ViT, and the `[cls]` is treated as the "sample centroid" refined from `[vis]`. Mathematically, the refined sample

centroid can be described as the weighted average of all visual tokens using a self-attention mechanism. Here, the sample centroid refers to the weighted aggregation of features from all visual tokens by the class token through a self-attention mechanism to form the global representation of the image. The concepts of **sample centroid manifold** and **class token manifold** are equivalent in nature.

Specifically, in the self-attention layer of the Transformer, each token (including `[cls]` and `[vis]`) calculates attention scores with respect to all other tokens. These attention scores are used to weight the features of each visual token for updating the class token. The self-attention mechanism can be represented as $\text{Attention}(\mathbf{q}, \mathbf{k}, \mathbf{v}) = \text{softmax}\left(\frac{\mathbf{q}\mathbf{k}^\top}{\sqrt{d_k}}\right)\mathbf{v}$. The $\mathbf{q}, \mathbf{k}, \mathbf{v}$ represent the query, key, and value matrices, respectively. These matrices are generated from the embedding vectors of tokens through linear layers. $d_k$ is the square root of the dimension of the key vectors. It is used to scale the dot products to prevent gradient vanishing or exploding.

For the class token, its update can be represented as:

$$[\text{cls}]' = \text{Attention}([\text{cls}], \mathbf{k}, \mathbf{v}) + [\text{cls}], \tag{1}$$

where $[\text{cls}]'$ represents the updated class token embedding, and $+$ denotes the residual connection. In the self-attention mechanism, the update of the class token can be seen as the weighted average of the features of all patch tokens, where the attention scores determine the weights:

$$[\text{cls}]' = \sum_{i=1}^{H \times W} \alpha_i [\text{vis}]_i + [\text{cls}]. \tag{2}$$

The $\alpha_i$ represents the attention score of the class token to the $i$-th patch token and $[\text{vis}]_i$ denotes the embedding vector of the $i$-th patch token. The class token can be regarded as the weighted average of the features of all patch tokens, known as the "sample centroid," where the self-attention mechanism dynamically computes the weights. This weighted average allows the class token to capture the global features of the image, rather than just a simple arithmetic mean.

Given $[\text{vis}]$ and $[\text{cls}]$, the extent of the sample centroid manifold, also known as the class token manifold extent (CTME), can be represented as:

$$CTME = \|[\text{cls}]\|_*, \tag{3}$$

where $\|\cdot\|_*$ represents the nuclear norm. The sample centroid manifold contains the magnitudes of each individual visual/patch token manifold. If Equation 3 is considered as the optimization objective, that is, when the sample centroid manifold is maximized, it implicitly minimizes each $[\text{vis}]$ manifold, thereby enhancing the intra-manifold similarity. Further understanding is provided in Subsection 3.2.

## 3.2 MAXIMUM CLASS TOKEN MANIFOLD CAPACITY

This subsection provides a detailed description of Maximum Class Token Manifold Capacity. Specifically, for the labeled samples provided in the GCD task, we assume that the annotations provided by human annotators are sufficiently accurate and unbiased. Therefore, supervised methods can effectively shape the manifold of these samples. As a result, we focus on enhancing the manifold capacity of the unlabeled samples.

The functions $f(\cdot)$ and $g(\cdot)$ perform as the feature extractor and projection head, respectively. Both the feature $\mathbf{h}_i = f(\mathbf{x}_i)$ and the projected embedding $\mathbf{z}_i = g(\mathbf{h}_i)$ are under L-2 normalization.

For the unlabeled samples in the mini-batch $\mathcal{B}^u$, after the feature extractor cuts them into $H \times W$ patches, the features are sent to the projection layer to obtain embeddings, which are the visual tokens of unlabeled samples:

$$[\text{vis}]^u = \mathbf{z}_i^u \overset{\text{def}}{=} g(f(\mathbf{x}_i^u)) \in \mathcal{Z}, \tag{4}$$

where, $\mathcal{Z}$ is commonly the $D$-dimensional hypersphere $\mathbb{S}^{D-1} \stackrel{\text{def}}{=} \{\mathbf{z} \in \mathbb{R}^D : \mathbf{z}^T\mathbf{z} = 1\}$ or $\mathbb{R}^D$.

Furthermore, from Equation 2, we can obtain the refined sample centroid that represents $[\texttt{vis}]^u$, which is denoted as $[\texttt{cls}]^u$, and define the loss function for maximum class token manifold capacity:

$$\mathcal{L}_{\text{MTMC}} \stackrel{\text{def}}{=} -\|[\texttt{cls}]^u\|_* \stackrel{\text{def}}{=} -\sum_{r=1}^{\text{rank}([\texttt{cls}]^u)} \sigma_r([\texttt{cls}]^u), \tag{5}$$

where $\sigma_r([\texttt{cls}]^u)$ is the $r$-th singular value of $[\texttt{cls}]^u$.

Minimizing the MTMC loss implies maximizing the nuclear norm of the class token. This means that without MTMC, the manifold of samples within clusters has a larger range, resulting in a lower nuclear norm of the centroid matrix. After training, in a geometric intuitive explanation, the $[\texttt{vis}]$ manifolds can be imagined as subspaces in a high-dimensional space, and each $[\texttt{vis}]$ manifold represents the possible value range of the corresponding slice feature. When maximizing CTME, geometrically speaking, $[\texttt{cls}]$ tries to find the most representative "center" position in the overall space composed of these $[\texttt{vis}]$ manifolds, so that a certain comprehensive distance (reflected in the nuclear norm) from all $[\texttt{vis}]$ to this "center" is minimized. As a result, the centroid matrix has a larger nuclear norm, and the representation within the cluster becomes more complete as the collapsed representations unravel.

As shown in the following code, the implementation of MTMC is extremely concise. The core code consists of only three lines. After calculating the loss $\mathcal{L}_{\text{GCD}}$ of any GCD scheme, the class token is obtained and singular value decomposition is performed, and the sum of singular values is added to the backward propagation of the loss $\mathcal{L}_{\text{GCD}} + \lambda\mathcal{L}_{\text{MTMC}}$.

```python
def forward(self, x_unlabel, loss):
    f_unlabel = self.featurizer(x_unlabel) # get class and visual tokens
    f_cls_unlabel = f_unlabel[:,0] # get class token
    z_cls_unlabel = self.projector(f_cls_unlabel) # get feature embedding
    _,s,_ = torch.svd(z_cls_unlabel) # singular value decomposition
    loss += self.lambda * torch.sum(s) # MTMC
    return loss
```

### 3.3 MAXIMUM CLASS TOKEN MANIFOLD CAPACITY INCREASES VON NEUMANN ENTROPY

The autocorrelation matrix of the test sample class token manifold is denoted as $\mathcal{A} \triangleq \sum_{i=1}^{N} \frac{1}{N}[\texttt{cls}]_i[\texttt{cls}]_i^\top = \mathbf{CLS}^\top\mathbf{CLS}/N$. We employ von Neumann entropy (Petz, 2001; Boes et al., 2019) to measure manifold capacity. This gives the advantage of focusing exclusively on the eigenvalues obtained after decomposition, allowing for graceful handling of eigenvalues that are extremely close to zero. The von Neumann entropy can be expressed as $\hat{H}(\mathcal{A}) \triangleq -\sum_j \lambda_j \log \lambda_j$, representing the Shannon entropy of the eigenvalues of $\mathcal{A}$, with values ranging between $0$ and $\log d$. A larger $\hat{H}(\mathcal{A})$ indicates a greater manifold capacity of the features.

Von Neumann entropy is an effective measure for assessing the uniformity of distributions and managing extreme values. As illustrated in Figure 2, the incorporation of MTMC results in a von Neumann entropy for the feature embeddings that is significantly higher than that of the original scheme. Furthermore, it is possible to relate von Neumann entropy to the rank of the $[\texttt{cls}]$. When $\mathcal{A}$ possesses uniformly distributed eigenvalues with full rank, the entropy is maximized, which can be explicitly expressed as below.

**Theorem 1** *For a given $[\texttt{cls}]$ autocorrelation $\mathcal{A} = \mathbf{CLS}^\top\mathbf{CLS}/N \in \mathbb{R}^{d \times d}$ of rank $k$ ($\leq d$),*

$$\log(\text{rank}(\mathcal{A})) \geq \hat{H}(\mathcal{A}) \tag{6}$$

*where equality holds if the eigenvalues of $\mathcal{A}$ are uniformly distributed with $\forall_{j=1}^{k}\lambda_j = 1/k$ and $\forall_{j=k+1}^{d}\lambda_j = 0$.*

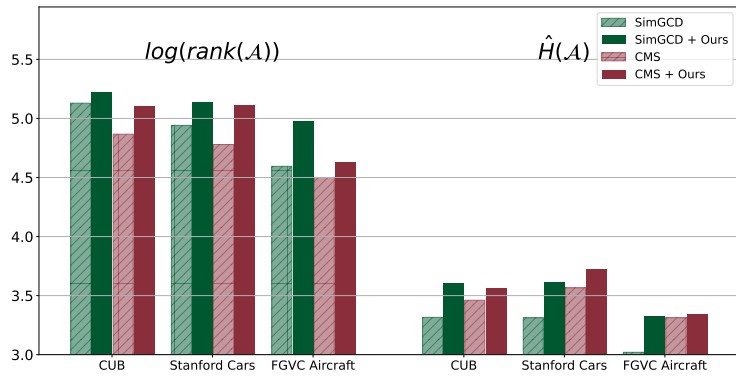

Figure 2: Comparison between $log(\text{rank}(\mathcal{A}))$ and $\hat{H}(\mathcal{A})$. The count of the largest eigenvalues necessary to account for 99% of the total eigenvalue energy serves as a surrogate for the rank.

A higher von Neumann entropy generally implies a larger manifold capacity. We provide a comparison of the von Neumann entropy for different schemes in Figure 2, and it can be clearly observed that MTMC has a higher value, indicating the high-rank nature of the features and the uniformity of neuron activation in each dimension of representation.

## 4 EXPERIMENTS

### 4.1 SETUP

**Benchmarks**. MTMC is evaluated on a total of six image recognition benchmarks. These include two conventional datasets, CIFAR100 (Krizhevsky et al., 2009) and ImageNet100 (Geirhos et al., 2019), and four fine-grained datasets, CUB-200-2011 (Wah et al., 2011), Stanford Cars (Krause et al., 2013), FGVC Aircraft (Maji et al., 2013), and Herbarium19 (Tan et al., 2019). To segregate target classes into sets of known and unknown, we adhere to the splits defined by the Semantic Shift Benchmark (SSB) (Vaze et al., 2021) when working with CUB, Stanford Cars, and FGVC Aircraft. The splits from the previous study (Vaze et al., 2022) is employed for the remaining datasets, we designate 80% of the classes as known under the CIFAR100 benchmark. For the rest of the benchmarks, the proportion of known classes stands at 50%. Our labeled set, known as $\mathcal{D}_l$, comprises 50% images from the known classes for all benchmarks.

**Evaluation Protocols**. We assess MTMC's effectiveness via a two-step process. First, we cluster the complete collection of images defined as $\mathcal{D}$. Then, we measure the accuracy on the set $\mathcal{D}_u$. In line with previous research (Vaze et al., 2022), accuracy is determined by comparing the assignments to the actual labels using the Hungarian optimal matching (Kuhn, 1955). This method bases the match on the number of instances that intersect between each pair of classes. Instances that do not belong to any pair, i.e., unpaired classes, are viewed as incorrect predictions. On the other hand, instances belonging to the most abundant class within each ground-truth cluster are taken as correct for accuracy calculations. We present the accuracy for all unlabeled data, and the accuracy is classified as old/known and new/novel, respectively. The accuracy using the estimated number of classes and the ground-truth $K$ are reported. This allows us to compare MTMC with previous studies that have assumed the availability of the $K$ during the evaluation phase.

**Implementation Details**. The purpose of MTMC is to empower existing GCD schemes to improve the completeness of representation. We closely adhere to their initial implementation details for an effective comparison. We use a pre-trained DINO ViT-B/16 (Caron et al., 2021; Dosovitskiy, 2020), utilizing it as our image encoder along with a projection head, an approach consistent with existing methods (Vaze et al., 2022; Zhang et al., 2023; Pu et al., 2023). The projection head consists of three 2,048-dimensional linear layers succeeded by GeLU activation. Only the parameters of the last layer of DINO and the projection head undergo training, while others are frozen. The dimension $D$ of the projection head is 768. All of our experiments are performed with a single NVIDIA RTX4090. The SGD optimizer (Ruder, 2016) is used with a batch size of 128 and a weight decay of 5e-5. The

Table 1: Comparison with the SOTAs on GCD, evaluated *with* or *without* the $K$ for clustering.

| Method | CIFAR100 | | | ImageNet100 | | | CUB | | | Stanford Cars | | | FGVC Aircraft | | | Herbarium 19 | | |
|---|---|---|---|---|---|---|---|---|---|---|---|---|---|---|---|---|---|---|
| | All | Old | New | All | Old | New | All | Old | New | All | Old | New | All | Old | New | All | Old | New |
| *(a) Clustering with the ground-truth number of classes $K$ given* († denotes reproduced results) | | | | | | | | | | | | | | | | | | |
| Agglomerative (Ward Jr, 1963) | 56.9 | 56.6 | 57.5 | 73.1 | 77.9 | 70.6 | 37.0 | 36.2 | 37.3 | 12.5 | 14.1 | 11.7 | 15.5 | 12.9 | 16.9 | 14.4 | 14.6 | 14.4 |
| RankStats+ (Han et al., 2020) | 58.2 | 77.6 | 19.3 | 37.1 | 61.6 | 24.8 | 33.3 | 51.6 | 24.2 | 28.3 | 61.8 | 12.1 | 26.9 | 36.4 | 22.2 | 27.9 | 55.8 | 12.8 |
| UNO+ (Fini et al., 2021) | 69.5 | 80.6 | 47.2 | 70.3 | 95.0 | 57.9 | 35.1 | 49.0 | 28.1 | 35.5 | 70.5 | 18.6 | 40.3 | 56.4 | 32.2 | 28.3 | 53.7 | 14.7 |
| ORCA (Cao et al., 2022) | 69.0 | 77.4 | 52.0 | 73.5 | 92.6 | 63.9 | 35.3 | 45.6 | 30.2 | 23.5 | 50.1 | 10.7 | 22.0 | 31.8 | 17.1 | 20.9 | 30.9 | 15.5 |
| GCD (Vaze et al., 2022) | 73.0 | 76.2 | 66.5 | 74.1 | 89.8 | 66.3 | 51.3 | 56.6 | 48.7 | 39.0 | 57.6 | 29.9 | 45.0 | 41.1 | 46.9 | 35.4 | 51.0 | 27.0 |
| DCCL (Pu et al., 2023) | 75.3 | 76.8 | 70.2 | 80.5 | 90.5 | 76.2 | 63.5 | 60.8 | 64.9 | 43.1 | 55.7 | 36.2 | - | - | - | - | - | - |
| PrCAL (Zhang et al., 2023) | **81.2** | 84.2 | 75.3 | 83.1 | 92.7 | 78.3 | 62.9 | 64.4 | 62.1 | 50.2 | 70.1 | 40.6 | 52.2 | 52.2 | 52.3 | 37.0 | 52.0 | 28.9 |
| GPC (Zhao et al., 2023) | 77.9 | 85.0 | 63.0 | 76.9 | 94.3 | 71.0 | 55.4 | 58.2 | 53.1 | 42.8 | 59.2 | 32.8 | 46.3 | 42.5 | 47.9 | - | - | - |
| PIM (Chiaroni et al., 2023) | 78.3 | 84.2 | 66.5 | 83.1 | 95.3 | 77.0 | 62.7 | **75.7** | 56.2 | 43.1 | 66.9 | 31.6 | - | - | - | 42.3 | 56.1 | 34.8 |
| SimGCD (Wen et al., 2023)† | 80.1 | 81.5 | 77.2 | 83.3 | 92.1 | 78.9 | 60.7 | 65.6 | 57.7 | 51.2 | 69.4 | 42.4 | 51.2 | 56.5 | 48.6 | 44.7 | 57.4 | 37.9 |
| + Ours | 80.2 | 81.5 | **77.5** | **86.7** | 93.1 | **83.6** | 62.1 | 65.8 | 60.3 | 52.3 | 70.0 | **43.7** | **55.1** | 58.9 | **53.1** | **45.6** | 57.8 | **39.0** |
| △ | +0.1 | +0.0 | +0.3 | +3.4 | +1.0 | +4.7 | +1.4 | +0.2 | +2.6 | +1.1 | +0.6 | +1.3 | +3.9 | +2.4 | +4.5 | +0.9 | +0.4 | +1.1 |
| CMS (Choi et al., 2024)† | 79.5 | 85.4 | 67.7 | 83.0 | **95.6** | 76.6 | 67.1 | 74.9 | 63.2 | 51.1 | **75.1** | 39.5 | 51.8 | **62.5** | 46.5 | 36.5 | 55.4 | 26.4 |
| + Ours | 79.0 | **85.5** | 66.1 | 84.8 | 95.6 | 79.5 | **71.1** | 74.1 | **66.9** | 52.5 | 73.9 | 42.1 | 52.0 | 61.8 | 47.0 | 36.3 | 56.5 | 25.4 |
| △ | -0.5 | +0.1 | -1.6 | +1.8 | +0.0 | +2.9 | +4.0 | -0.8 | +3.7 | +1.4 | -1.2 | +2.6 | +0.2 | -0.7 | +0.5 | -0.2 | +1.1 | -1.0 |
| *(b) Clustering without the ground-truth number of classes $K$ given* | | | | | | | | | | | | | | | | | | |
| Agglomerative (Ward Jr, 1963) | 56.9 | 56.6 | 57.5 | 72.2 | 77.8 | 69.4 | 35.7 | 33.3 | 36.9 | 10.8 | 10.6 | 10.9 | 14.1 | 10.3 | 16.0 | 13.9 | 13.6 | 14.1 |
| GCD (Vaze et al., 2022) | 70.8 | 77.6 | 57.0 | 77.9 | 91.1 | 71.3 | 51.1 | 56.4 | 48.4 | 39.1 | 58.6 | 29.7 | - | - | - | 37.2 | 51.7 | 29.4 |
| GPC (Zhao et al., 2023) | 75.4 | 84.6 | 60.1 | 75.3 | 93.4 | 66.7 | 52.0 | 55.5 | 47.5 | 38.2 | 58.9 | 27.4 | 43.3 | 40.7 | 44.8 | 36.5 | 51.7 | 27.9 |
| PIM (Chiaroni et al., 2023) | 75.6 | 81.6 | 63.6 | 83.0 | 95.3 | 76.9 | 62.0 | **75.7** | 55.1 | 42.4 | 65.3 | 31.3 | - | - | - | **42.0** | 55.5 | **34.7** |
| CMS (Choi et al., 2024)† | 77.8 | 84.0 | 65.3 | 83.4 | 95.6 | 77.3 | 66.2 | 69.7 | 64.4 | 47.2 | 67.6 | 37.3 | 50.8 | **60.0** | 46.2 | 38.5 | **57.3** | 28.4 |
| + Ours | **79.5** | 84.7 | **69.1** | **84.3** | 95.7 | **78.8** | **68.7** | 74.1 | 66.0 | 50.6 | 70.3 | 41.0 | **51.1** | 57.7 | **47.7** | 38.0 | 56.9 | 27.9 |
| △ | +1.7 | +0.7 | +3.8 | +0.9 | +0.1 | +1.5 | +2.5 | +4.4 | +1.6 | +3.4 | +2.7 | +3.7 | +0.3 | -2.3 | +1.5 | -0.5 | -0.4 | -0.5 |

Table 2: Estimated number and error rate of $K$.

| Method | CIFAR100 | | ImageNet100 | | CUB | | Stanford Cars | | FGVC Aircraft | | Herbarium 19 | |
|---|---|---|---|---|---|---|---|---|---|---|---|---|
| | K | Err(%) | K | Err(%) | K | Err(%) | K | Err(%) | K | Err(%) | K | Err(%) |
| Ground truth | 100 | - | 100 | - | 200 | - | 196 | - | 100 | - | 683 | - |
| GCD (Vaze et al., 2022) | 100 | 0 | 109 | 9 | 231 | 15.5 | 230 | 17.3 | - | - | 520 | 23.8 |
| DCCL (Pu et al., 2023) | 146 | 46 | 129 | 29 | 172 | 9 | 192 | 0.02 | - | - | - | - |
| PIM (Chiaroni et al., 2023) | 95 | 5 | 102 | 2 | 227 | 13.5 | 169 | 13.8 | - | - | **563** | **17.6** |
| GPC (Zhao et al., 2023) | 100 | 0 | 103 | 3 | **212** | **6** | 201 | 0.03 | - | - | - | - |
| CMS (Choi et al., 2024)† | 94 | 6 | 98 | 2 | 176 | 12 | 149 | 23.9 | 88 | 12 | 503 | 26.4 |
| + Ours | **96** | **4** | **100** | **0** | 180 | 10 | **159** | **18.9** | 89 | 11 | 508 | 25.6 |

count of the largest eigenvalues necessary to account for 99% of the total eigenvalue energy serves as a surrogate for the rank in Equation 5.

## 4.2 MAIN RESULTS

**Evaluation on GCD.** Table 1 presents a comprehensive comparison of the results of GCD that can and cannot be obtained for the number of categories $K$ on coarse-grained and fine-grained datasets. The summary is as follows: (1) We conducted experiments with $\mathcal{L}_{\text{SimGCD}} + \mathcal{L}_{\text{MTMC}}$. A notable result is that although SimGCD has already achieved high accuracy, MTMC can still significantly enhance its performance ceiling, especially in the perception of novel classes. On ImageNet100, MTMC improved by 4.7%, advancing the model towards the real world. Even on challenging datasets like Herbarium19, there is a comprehensive improvement. (2) Under the optimization target of $\mathcal{L}_{\text{CMS}} + \mathcal{L}_{\text{MTMC}}$, MTMC has increased the accuracy rate for all categories, indicating that it has improved the representation quality of unknown classes without compressing the embedding space of known classes as much as possible. (3) A point worth noting is that the clustering effect of CMS+MTMC is better when without $K$ than with $K$. Known classes on the CUB and novel classes on CIFAR100 and Stanford Cars datasets have achieved nearly a 4% performance gain, which confirms the viewpoint of this paper that human intelligence-imparted category attributes are biased. When without $K$, MTMC stimulates the model's potential for perceiving the open world without the intervention of human-defined biased definitions.

**Estimated number of clusters.** We present the gap between MTMC and SOTAs in estimating the number of clusters in Table 2. Leveraging the CMS, which does not require specific hyper-

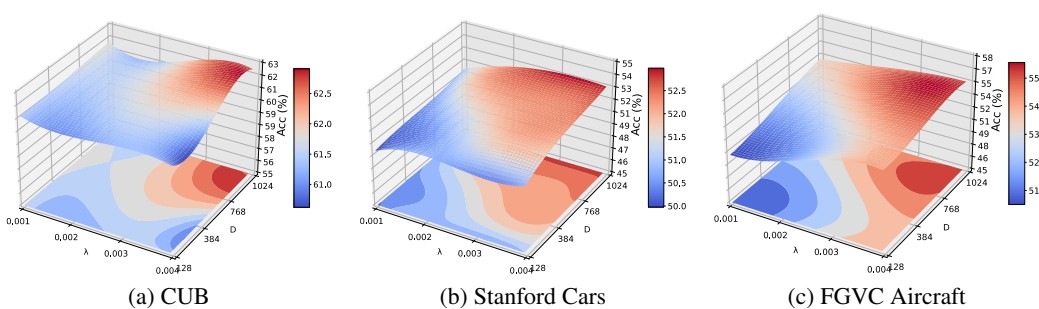

|     (a) CUB     |     (b) Stanford Cars     |     (c) FGVC Aircraft     |

Figure 3: Hyperparameter sensitivity of the degree of MTMC $\lambda$ and features dimensionality $D$.

parameters to estimate $K$, our optimization target is $\mathcal{L}_{\text{CMS}} + \mathcal{L}_{\text{MTMC}}$. The results show marked improvement when MTMC is incorporated into the CMS framework. This enhancement is significant and consistent across various datasets, showcasing the model's ability to separate different classes more accurately. Notably, on the ImageNet100 dataset, which is known for its complexity and diversity, our method achieves a remarkable 100% correct estimation rate. It symbolizes the model's advanced capability to discern fine-grained distinctions between classes, suggesting a high degree of alignment between the learned decision boundaries and the intrinsic structure of the data. The enhancement in correctly estimating the number of clusters underscores the importance of representation completeness. A richer and more complete representation within each class allows the model to capture better the nuances and variability that are characteristic of that class. This, in turn, sharpens the distinctions between different classes, leading to more precise and reliable inter-class separation. Moreover, an accurate estimation of $K$ indicates a model that is not only performing well in terms of clustering accuracy but is also aligned with the principles of real-world categorization. When the decision boundaries set by the model reflect the actual divisions in the data, it implies a deeper understanding of the underlying structure of the dataset. This alignment is crucial for applications where the number of potential categories is unknown or could change over time, such as in open-world learning scenarios.

**Ablation study.** The only hyperparameter of MTMC is the coefficient $\lambda$ of the loss. To gain a deeper understanding of the correlation between the degree of maximum token manifold capacity and the dimensionality $D$ of the features, we conducted an ablation experiment on it, as shown in Figure 3. It can be clearly observed that MTMC is not sensitive to hyperparameters and can uniformly enhance clustering accuracy. A more thought-provoking finding is that directly reducing $D$ to avoid dimensionality collapse is suboptimal. The reason is that each dimension of the manifold contributes to the representation, and a reduction in $D$ will directly lead to a loss of information. Even with MTMC, it is impossible to make the representation complete. An appropriate number of dimensions enriches the representation while using MTMC to prevent dimensionality collapse, which can maximize the model's performance enhancement.

### 4.3 ANALYTICAL RESULTS

**Impact of embedding quality.** In Table 1, the accuracy gains on the CIFAR100 and Herbarium19 datasets are insignificant. We use this as a starting point to analyze the conflict between enhancing feature completeness and low embedding quality in GCD. DINO, through self-supervision, already has a good feature representation capability, but due to the distribution of data, its embedding quality still be low. One source of low quality is the data size, and the other is data semantics. (1) Specifically, when the small-sized CIFAR10 images are interpolated and input into ViT, the high-frequency information is lost. For example, when identifying animal categories, the low-frequency features such as the outline of the animal may be captured relatively well, but the detailed features such as the texture and eyes of the animal (high-frequency features) are difficult to accurately extract. In this case, the model can only cluster through some shortcut information, rather than accurately clustering based on the complete intra-class features. Since the manifold dimension of the low-frequency features is relatively low, it is unable to fully capture the diversity and complexity within the class. Therefore, enhancing the completeness of the intra-class representation on small-

sized data is challenging. (2) Herbarium19 is a large-scale herbal plant recognition dataset, which is not in the model's training data and inherently cannot provide highly discriminative representations. Additionally, the large number of categories makes the decision boundary more chaotic, and existing GCD schemes cannot cluster well. Therefore, enhancing the completeness of intra-class representation on overly low-quality embeddings is not feasible, as the overlap of feature spaces across categories is too large, and samples within a cluster come from multiple categories.

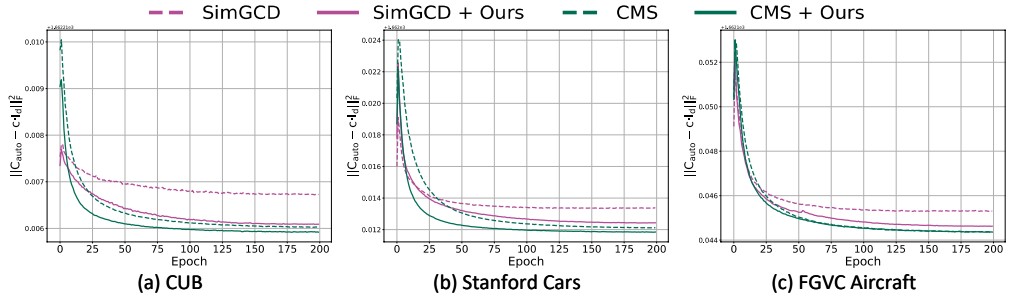

(a) CUB      (b) Stanford Cars      (c) FGVC Aircraft

Figure 4: The Frobenius norm $\|\mathcal{A} - c \cdot I_d\|_F^2$ on three fine-grained benchmarks.

**MTMC homogenizes eigenvalue distribution and reduces Frobenius norm.** The autocorrelation matrix of the test sample class token manifold is denoted as $\mathcal{A}$. Given that $\|[\texttt{cls}]_i\|_2 = 1$ and $\mathcal{A} \geq 0$, it can be easily verified that $\sum_j \lambda_j = 1$ and $\forall_j \lambda_j \geq 0$ (Parkhi et al., 2015; Liu et al., 2017; Mettes et al., 2019), where $\{\lambda_j\}$ is the eigenvalues of $\mathcal{A}$. Under the ideal condition where $\mathcal{A} \rightarrow c \cdot I_d$, which represents the maximum manifold capacity, the eigenvalue distribution of $\mathcal{A}$ becomes completely uniform, $\mathbf{z}$ becomes uncorrelated (Cogswell et al., 2015), full-rank (Hua et al., 2021), and isotropic (Vershynin, 2018). It can be seen that $\mathcal{A}$ is closely related to various characteristics of representation. Furthermore, the Frobenius norm (Ma et al., 1994; Peng et al., 2016), extensively studied in self-supervised learning methods (Cogswell et al., 2015; Xiong et al., 2016; Choi & Rhee, 2019; Zbontar et al., 2021), serves as a measure of whether the model output representation relies predominantly on a few neurons or dimensions (The Frobenius norm calculates the square root of the sum of the squares of all elements of the matrix, and it measures the "size" or "energy" of the matrix as a whole. When the Frobenius norm is small, it means that the overall "energy" of the matrix elements is relatively low. From the perspective of feature representation, this may indicate that the model does not overly rely on certain specific dimensions or feature combinations when extracting features). It also reflects the size of the manifold capacity. A smaller Frobenius Norm indicates a larger manifold capacity. We conducted singular value decomposition (SVD) (Golub & Reinsch, 1971) on the autocorrelation matrix of the feature embeddings derived from the test set, subsequently plotting the first 200 singular values in descending order, as shown in Figure 5 and We visualize the Frobenius norm $\|\mathcal{A} - c \cdot I_d\|_F^2$ in Figure 4. Compared to the original SimGCD and CMS, MTMC effectively achieves a more uniform eigenvalue distribution and significantly reduces the Frobenius norm.

**MTMC unravels dimensional collapse.** The completeness of features profoundly influences the richness of intra-class representations, thereby impacting clustering accuracy (Figure 5). Features characterized by high completeness also exhibit a substantial manifold capacity. It is evident that MTMC, which offers a greater manifold capacity, yields a higher mean of singular values. This observation implies that the tail singular values contribute significantly to the representation of samples. A richer representation facilitates clusters approximating the true uncompressed distribution, thereby enhancing clustering accuracy. Conversely, while CMS and SimGCD contribute to clustering, they operate within a lower-dimensional space, where only a limited number of singular values hold significance. This limitation reduces the manifold capacity, and the incomplete representation constrains the model's potential performance. The theory of dimension collapse (Caron et al., 2020; Shi et al., 2023) posits that the singular values of the covariance matrix of feature embeddings serve as critical indicators for assessing the severity of dimension collapse. While strong unconstrained contrastive learning facilitates compact clustering, it simultaneously leads to dimension collapse, resulting in a low-dimensional feature embedding space where an increasing number of singular values approach zero. From a modeling perspective, dimension collapse embodies a form of oversimplification, representing a shortcut that suggests the space has not been fully leveraged to distinguish

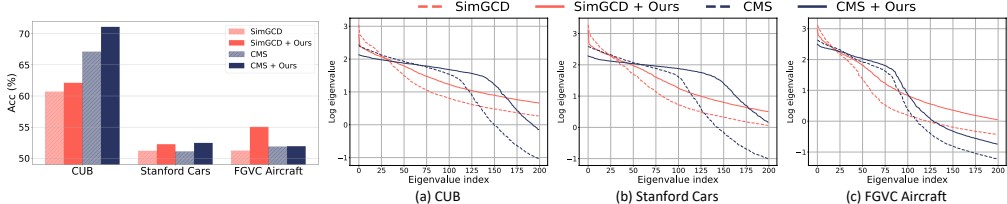

Figure 5: MTMC effectively mitigates dimensional collapse by providing a more uniform eigenvalue distribution and improves the clustering accuracy.

diverse samples within the same category. Unlike traditional methods, MTMC prioritizes maximizing the completeness of intra-class distributions rather than inter-class separation, thereby providing more precise decision boundaries.

## 5 RELATED WORKS

### 5.1 GENERALIZED CATEGORY DISCOVERY

Generalized category discovery (Vaze et al., 2022; Zhao et al., 2023; Wen et al., 2023; Choi et al., 2024) is crucial for identifying and classifying both known and new categories in a dataset, expanding beyond traditional supervised learning to recognize new classes not seen during training. The pioneering work (Vaze et al., 2022) establishes a framework that employs semi-supervised k-means clustering. Following this initial proposition, SimGCD (Wen et al., 2023) is introduced as a parametric classification approach that utilizes entropy regularization and self-distillation. Expanding on these concepts, CMS (Choi et al., 2024) is proposed, enhancing representation learning through mean-shift based clustering. Moreover, a deep clustering approach (Zhao et al., 2023) emerges that dynamically adjusts the number of prototypes during inference, facilitating an adaptive discovery of new categories. Most recently, ActiveGCD (Ma et al., 2024) actively selects samples from unlabeled data to query for labels, with the aim of enhancing the discovery of new categories through an adaptive sampling strategy. Each of these contributions addresses the multifaceted challenges of representation learning, category number estimation, and label assignment, redefining the frontiers of open-world learning. Regardless of the flourishing development of GCD, their focus remains on compact clustering, neglecting the integrity of intra-class representation. Our goal is to empower any GCD scheme with concise means to promote the non-collapse representation of each sample, thus shaping more accurate decision boundaries.

### 5.2 DIMENSIONAL COLLAPSE

This Dimensional collapse (Grill et al., 2020; Caron et al., 2020; Shi et al., 2023; Jing et al., 2021) occurs when the learned embeddings tend to concentrate within a lower-dimensional subspace rather than dispersing throughout the entire embedding space, thereby limiting the representations' capacity for diversity and expressiveness. DirectCLR (Jing et al., 2021) presents a direct optimization of the representation space, sidestepping the need for a trainable projector, which inherently mitigates the risk of dimensional collapse by promoting a more even distribution of embeddings across the space. Complementing this, the whitening approach (Tao et al., 2024) standardizes covariance matrices through whitening techniques, ensuring that each dimension contributes equally to the representation, thus preventing any subset of dimensions from dominating the learning process. Similarly, the non-contrastive learning objective (Chen et al., 2024) for collaborative filtering avoids data augmentation and negative sampling, focusing on alignment and compactness within the embedding space to prevent dimensional collapse. The Bregman matrix divergence (Zhang et al., 2024) further fortifies the fight against dimensional collapse by minimizing the distance between covariance matrices and the identity matrix, ensuring a uniform distribution of embeddings and directly countering the concentration of information along certain dimensions. Moreover, random orthogonal projection image modeling (Haghighat et al., 2023) provides a preventative measure against dimensional collapse by modeling images with random orthogonal projections, which promotes the exploration of a wide range of features and discourages the concentration on a limited subset of dimensions.

Rather than directly addressing the issue of dimensional collapse, we focus on maximizing token manifold capacity to align the radius and dimensions of the manifold with the rich distribution of the real world. This approach also unravels the sample-level dimensional collapse.

## 6 CONCLUSION

The paper introduces a straightforward approach to enhancing Generalized Category Discovery by Maximum Token Manifold Capacity. Our method counters the traditional focus on compact clusters, which can lead to low manifold capacity and incomplete representations. Emphasizing the integrity of intra-class representations, MTMC leverages the nuclear norm to ensure manifolds are both compact and informative. Through extensive experiments, we demonstrated that our proposal significantly improves clustering accuracy and the estimation of category numbers. Theoretically, MTMC prevents dimensional collapse, leading to a more uniform eigenvalue distribution and higher entropy, indicative of richer representations. Our method's effectiveness in GCD lies in its promotion of complete and non-collapsed representations, paving the way for more robust and adaptable machine learning models in open-world scenarios.

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
