# MTMC: Generalized Category Discovery via Maximum Token Manifold Capacity

# (Supplemental Material)

This document provides more details of our approach and additional experimental results, organized as follows:

## 1 DETAILS OF OPTIMIZATION OBJECTIVE OF GCD

The existing GCD proposals are all proposed for compact clustering. Summarizing the optimization objectives of mainstream schemes GCD (Vaze et al., 2022), CMS (Choi et al., 2024) and SimGCD (Wen et al., 2023), it can be observed that they are based on contrastive learning or prototype learning to significantly reduce the distance between potentially similar samples in the feature space.

### 1.1 GCD

The pioneering work (Vaze et al., 2022) divided the mini-batch $\mathcal{B}$ into labelled $\mathcal{B}^l$ and unlabeled $\mathcal{B}^u$, using supervised (Khosla et al., 2020) contrastive learning $\mathcal{L}_{\text{GCD}}^l = -\frac{1}{|\mathcal{B}^l|} \sum_{i \in \mathcal{B}^l} \frac{1}{|\mathcal{B}^l(i)|} \sum_{j \in \mathcal{B}^l(i)} \log \frac{\exp(\mathbf{z}_i^\top \mathbf{z}_j'/\tau)}{\sum_{n \neq i} \exp(\mathbf{z}_i^\top \mathbf{z}_n'/\tau)}$, and self-supervised (Chen et al., 2020) contrastive learning $\mathcal{L}_{\text{GCD}}^u = -\frac{1}{|\mathcal{B}|} \sum_{i \in \mathcal{B}} \log \frac{\exp(\mathbf{z}_i^\top \mathbf{z}_i'/\tau)}{\sum_{n \neq i} \exp(\mathbf{z}_i^\top \mathbf{z}_n'/\tau)}$ and balancing them using coefficients $\lambda$: $\mathcal{L}_{\text{GCD}} = (1-\lambda)\mathcal{L}_{\text{GCD}}^u + \lambda \mathcal{L}_{\text{GCD}}^l$, where $\mathcal{B}^l(i)$ represents the collection of samples with the same label as $i$. The $\mathbf{z}$ and $\mathbf{z}'$ are augmented from two different views, and the $\tau$ is the temperature.

### 1.2 CMS

CMS (Choi et al., 2024) and GCD adopt similar supervised and self-supervised contrastive learning. The difference is that CMS introduced mean-shift into unsupervised learning. For the $i$-th sample, CMS collects the feature set $\mathcal{V} = \{\mathbf{z}_i\}_{i=1}^N$ of training samples and calculates the k-nearest neighbours $\mathcal{N}(\mathbf{z}_i) = \{\mathbf{z}_i\} \cup \text{argmax}_{\mathbf{z}_j \in \mathcal{V}}^k \mathbf{z}_i \cdot \mathbf{z}_j$, where $\text{argmax}_{s \in \mathcal{S}}^k(\cdot)$ returns a subset of the top-$k$ items. By aggregating neighbor embeddings with weight kernel $\varphi(\cdot)$, it obtains the new embedded representation of samples after mean-shift: $\hat{\mathbf{z}}_i = \frac{\sum_{\mathbf{z}_j \in \mathcal{N}(\mathbf{z}_i)} \varphi(\mathbf{z}_j - \mathbf{z}_i)\mathbf{z}_j}{\left\| \sum_{\mathbf{z}_j \in \mathcal{N}(\mathbf{z}_i)} \varphi(\mathbf{z}_j - \mathbf{z}_i)\mathbf{z}_j \right\|}$. $\mathcal{L}_{\text{CMS}}$ and $\mathcal{L}_{\text{GCD}}$ are formally approximate.

### 1.3 SimGCD

SimGCD (Wen et al., 2023) constructs a prototype classifier $\mathbf{C} = \{\mathbf{c}_1, \cdots, \mathbf{c}_{K_{\text{known}}+K_{\text{novel}}}\}$ for both known and unknown classes. It obtains the posterior probability $\mathbf{p}_i^{(k)} = \frac{\exp(\mathbf{h}_i^\top \mathbf{c}_k)/\tau}{\sum_{k'} \exp(\mathbf{h}_i^\top \mathbf{c}_k')/\tau}$ in a similar way to FixMatch and uses cross-entropy loss $\mathcal{L}_{\text{SimGCD}}^l = \frac{1}{|\mathcal{B}^l|} \sum_{i \in \mathcal{B}^l} \ell(y_i, \mathbf{p}_i)$ on labeled samples. Self-distillation and entropy regularization $\mathcal{L}_{\text{SimGCD}}^u = \frac{1}{|\mathcal{B}|} \ell(\mathbf{p}_i', \mathbf{p}_i) - \lambda_e H(\frac{1}{2|\mathcal{B}|} \sum_{i \in \mathcal{B}} (\mathbf{p}_i + \mathbf{p}_i'))$ are performed using augmented samples with probability $\mathbf{p}_i'$.

## 2 DETAILS OF MAXIMUM MANIFOLD CAPACITY REPRESENTATION

Manifold Capacity Theory (Yerxa et al., 2023; Schaeffer et al., 2024; Isik et al., 2023) is a theoretical framework used to evaluate the efficiency of neural representation coding. Its core idea is to map the complex structures in high-dimensional datasets to low-dimensional manifolds. These manifolds represent different objects or categories in the feature space. The following formulas can illustrate some key concepts in Manifold Capacity Theory:

### 2.1 MANIFOLD RADIUS

$R_M = \sqrt{\frac{1}{P} \sum_{i=1}^{P} \lambda_i^2}$. $\lambda_i$ represents the eigenvalues of the covariance matrix of points on the manifold, and $P$ denotes the number of points on the manifold. The manifold radius measures the size of the manifold relative to its centroid.

### 2.2 MANIFOLD DIMENSIONALITY

$D_M = \frac{\left(\sum_{i-1}^{P} \lambda_i\right)^2}{\sum_{i-1}^{P} \lambda_i^2}$. This is a general participation rate that quantifies the extent of expansion of the manifold along its major directions.

### 2.3 MANIFOLD CAPACITY

$\alpha_C = \phi\left(R_M \sqrt{D_M}\right)$. $\phi(\cdot)$ is a monotonically decreasing function with respect to its parameters. The manifold capacity $\alpha_C$ can be understood as the maximum number of manifolds that can be linearly separable in a given feature space.

Manifold capacity is evaluated based on the manifold radius and dimension to determine the maximum number of distinguishable categories in high-dimensional space. Optimizing Manifold Capacity aims to improve the coding efficiency of representations by optimizing the geometric properties of the manifold. Formally, it can be represented as $\mathbf{Z}^* = \arg\min_{\mathbf{Z}} \|\mathbf{G}\mathbf{Z}\|_*$, where $\mathbf{G}$ is a symmetric matrix that encodes the semantic relationship between different augmented views, and $\mathbf{Z}$ is an embedding matrix that contains the representation vectors of all augmented views. The optimal embedding $\mathbf{Z}^*$ is the one that minimizes the nuclear norm of $\mathbf{G}\mathbf{Z}$.

## 3 THEORETICALLY NECESSARY OF MTMC

GCD is a semi-supervised learning scheme and MTMC is theoretically necessary for GCD. We conduct a comprehensive analysis and derivation from High-Dimensional Probability perspectives (with special consideration given to the more general cases where the number of points $P$ is large and the dimension $D$ is high):

GCD aims to cluster the embeddings of samples from the same category as closely as possible, regardless of whether they are from known or unknown classes. That is each cluster center lies on the hypersphere, and the distribution of the centers on the hypersphere is made as uniform as possible. More formally, this goal can be replaced by two definitions in previous studies (Gálvez et al., 2023; Wang & Isola, 2020):

**Definition 1** (*Perfect Reconstruction*). A network $f_\theta$ is Perfect Reconstruction if $\forall \mathbf{x} \in \mathcal{X}, \forall t^{(1)}, t^{(2)} \in \mathcal{T}, \mathbf{z}^{(1)} = f_\theta\left(t^{(1)}(\mathbf{x})\right) = f_\theta\left(t^{(2)}(\mathbf{x})\right) = \mathbf{z}^{(2)}$, where $\mathcal{T}$ is a set of data aug-

mentations such as color jittering, cropping, flipping, etc. The dataset is $\mathbf{x}_{1:P}$ with $P$ samples, and the set after applying $1:K$ augmentation methods is $t^{(1)}(\mathbf{x}_p), \ldots, t^{(K)}(\mathbf{x}_p)$.

**Definition 2** (*Perfect Uniformity*). $p(Z)$ is the distribution over the network representations induced by the data and transformation sampling distributions. If $p(Z)$ is a uniform distribution on the hypersphere, then the network $f_\theta$ achieves Perfect Uniformity.

Intuitively, perfect reconstruction means that the network maps all views of the same data to the same embedding, while perfect uniformity means that these embeddings are uniformly distributed on the hypersphere. For brevity, we denote the centroid embedding of the class token representing the $p$-th sample under different augmentations as $\mathbf{z}_p$. We prove the following: A network that simultaneously achieves perfect reconstruction and perfect uniformity achieves a lower bound of what MTMC has, that is, it provides the lowest probability of $\mathcal{L}_{MTMC}$.

**Proposition 1**. *Suppose that,* $\forall p \in [P], \mathbf{c}_p^T \mathbf{c}_p \leq 1$. *Then,* $0 \leq \|C\|_* \leq \sqrt{P \min(P, D)}$.

*Proof.* Let $\sigma_1, \ldots, \sigma_{\min(P,D)}$ denote the singular values of $C$, so that $\|C\|_* = \sum_{i=1}^{\min(P,D)} \sigma_i$. The lower bound follows by the fact that singular values are nonnegative. For the upper bound, we have

$$\sum_{i=1}^{\min(P,D)} \sigma_i^2 = \mathrm{Tr}\left[CC^T\right] = \sum_{n=1}^{P} \mathbf{c}_p^T \mathbf{c}_p \leq P \tag{1}$$

Then, by Cauchy-Schwarz on the sequences $(1, \ldots, 1)$ and $\left(\sigma_1, \ldots, \sigma_{\min(P,D)}\right)$, we get

$$\sum_{i=1}^{\min(P,D)} \sigma_i \leq \sqrt{\left(\sum_{i=1}^{\min(P,D)} 1\right)\left(\sum_{i=1}^{\min(P,D)} \sigma_i^2\right)} \leq \sqrt{\min(P, D)P}. \tag{2}$$

**Proposition 2**. Let $f_\theta$ achieve perfect reconstruction. Then, $\|\mathbf{c}_p\|_2 = 1 \forall n$.

*Proof.* Because $f_\theta$ achieves perfect reconstruction, $\forall n, \forall t^{(1)}, t^{(2)}, \mathbf{z}_p^{(1)} = \mathbf{z}_p^{(2)}$. Thus $\mathbf{c}_p = (1/K)\sum_k \mathbf{z}_p^{(k)} = (1/K)\sum_k \mathbf{z}_p^{(1)} = \mathbf{z}_p^{(1)}$, and since $\left\|\mathbf{z}_p^{(1)}\right\|_2 = 1$, we have $\|\mathbf{c}_p\|_2 = 1$.

**Theorem 1**. Let $f_\theta : \mathcal{X} \rightarrow \mathbb{S}^D$ be a network that achieves perfect reconstruction and perfect uniformity. Then $f_\theta$ achieves the lower bound of $\mathcal{L}_{MTMC}$ with high probability. Specifically:

$$\|C\|_* = \begin{cases} P(1 - O(P/D)) & \text{if } P \leq D \\ \sqrt{PD}(1 - O(D/P)) & \text{if } P \geq D \end{cases} \tag{3}$$

with high probability in $\min(P, D)$.

This demonstrates that the MTMC loss can be minimized by minimizing the distances of all embeddings corresponding to the same datum and maximizing the distances of all samples' centers.

The above derivations and analyses based on High-Dimensional Probability demonstrate, the **theoretical strong correlation** of MTMC and GCD (as a type of semi-supervised learning).

## 4 PROOFS OF THEOREM

**Lemma 1** *Given non-negative values* $p_i$ *such that* $\sum_{i=1}^{n} p_i = 1$, *the entropy function* $H(p_1, \ldots, p_n) = -\sum_{i=1}^{n} p_i \log p_i$ *is strictly concave. Furthermore, it is upper-bounded by* $\log n$, *as demonstrated by the inequality,*

$$\log n = H(1/n, ..., 1/n) \geq H(p_1, ..., p_n) \geq 0. \tag{4}$$

**Proof 4.1** *Refer to Section D.1 in (Marshall, 1979).*

**Lemma 2** *The Kullback-Leibler (KL) divergence between two zero-mean, $d$-dimensional multivariate Gaussian distributions can be formulated as follows,*

$$
\begin{aligned}
&D_{\mathrm{KL}}(\mathcal{N}(0, \boldsymbol{\Sigma}_1) \| \mathcal{N}(0, \boldsymbol{\Sigma}_2)) \\
&= \frac{1}{2} \left[ tr(\boldsymbol{\Sigma}_2^{-1} \boldsymbol{\Sigma}_1) - d + \log \frac{|\boldsymbol{\Sigma}_2|}{|\boldsymbol{\Sigma}_1|} \right].
\end{aligned}
\tag{5}
$$

**Proof 4.2** *Refer to Section 9 in (Duchi, 2007).*

**Theorem 1** *For a given* `[cls]` *autocorrelation $\mathcal{A} = \mathbf{CLS}^\top \mathbf{CLS}/N \in \mathbb{R}^{d \times d}$ of rank $k$ ($\leq d$),*

$$
\log(\mathrm{rank}(\mathcal{A})) \geq \hat{H}(\mathcal{A})
\tag{6}
$$

*where equality holds if the eigenvalues of $\mathcal{A}$ are uniformly distributed with $\forall_{j=1}^k \lambda_j = 1/k$ and $\forall_{j=k+1}^d \lambda_j = 0$.*

**Proof 4.3**

$$
\begin{aligned}
log(rank(\mathcal{A})) &= log(k) &\text{(7)}\\
&\geq H(\lambda_1, ..., \lambda_k) \text{ (by Lemma 1)} &\text{(8)}\\
&= -\sum_{j=1}^k \lambda_j \log \lambda_j &\text{(9)}\\
&= -\sum_{j=1}^d \lambda_j \log \lambda_j &\text{(10)}\\
&= \hat{H}(\mathcal{A}). &\text{(11)}
\end{aligned}
$$

*According to Lemma 1, the inequality equation 8 attains equality if and only if $\lambda_j = \frac{1}{k}$ for all $j = 1, 2, \ldots, k$. Equation equation 10 adheres to the convention that $0 \log 0 = 0$, as per the definition in (Thomas & Joy, 2006).*

More details about the definition of $\lambda = 1$. Suppose we have a set of $n$ normalized vectors $\mathbf{v}_1, \mathbf{v}_2, \ldots, \mathbf{v}_n$, where the second-order norm (or length) of each vector is 1, that is, $\|\mathbf{v}_i\| = 1$ for all $i$. The autocorrelation matrix $\mathbf{A}$ of these vectors is defined as:

$$
\mathbf{A} = \frac{1}{n} \sum_{i=1}^n \mathbf{v}_i \mathbf{v}_i^T
\tag{12}
$$

Here, $\mathbf{v}_i \mathbf{v}_i^T$ is the outer product of the vector $\mathbf{v}_i$ with itself, which is a rank-1 matrix. The autocorrelation matrix $\mathbf{A}$ is the average of these outer product matrices.

Next, we need to find the eigenvalues of $\mathbf{A}$. Since $\mathbf{v}_i$ is normalized, $\mathbf{v}_i^T \mathbf{v}_i = 1$. This means that each $\mathbf{v}_i$ is an eigenvector of $\mathbf{A}$ with the corresponding eigenvalue of $\frac{1}{n}$. This is because:

$$
\mathbf{A}\mathbf{v}_i = \frac{1}{n} \left( \sum_{j=1}^n \mathbf{v}_j \mathbf{v}_j^T \right) \mathbf{v}_i = \frac{1}{n} \sum_{j=1}^n \mathbf{v}_j (\mathbf{v}_j^T \mathbf{v}_i) = \frac{1}{n} \mathbf{v}_i (\mathbf{v}_i^T \mathbf{v}_i) = \frac{1}{n} \mathbf{v}_i \cdot 1 = \frac{1}{n} \mathbf{v}_i
\tag{13}
$$

So, the eigenvalue corresponding to each $\mathbf{v}_i$ is $\frac{1}{n}$. Since $\mathbf{A}$ is a rank-$n$ matrix (assuming the vectors $\mathbf{v}_1, \mathbf{v}_2, \ldots, \mathbf{v}_n$ are linearly independent), it has $n$ eigenvalues. We already know that $n$ of the $n$ eigenvalues are $\frac{1}{n}$. Therefore, the sum of all the eigenvalues of $\mathbf{A}$ is:

$$
\text{sum of eigenvalues} = \frac{1}{n} + \frac{1}{n} + \cdots + \frac{1}{n} = n \cdot \frac{1}{n} = 1
\tag{14}
$$

# 5 MORE ANALYSIS

## 5.1 VISUALIZATION OF T-SNE

By maximizing the manifold capacity of the cls token, MTMC effectively enhances the high-dimensional feature representations, making them more concrete and uniform within each category. This approach mitigates potential dimensional collapse issues and improves the consistency of category samples in feature space distribution. This enhancement is evident in the T-SNE visualization of feature distribution (Figure 1), where the feature clustering using the embedded MTMC method preserves consistent distances between category centers while ensuring a more uniform size for each cluster. Consequently, the sample representations extracted by the model do not exhibit abrupt divergences among categories due to the influence of ground truth labels.

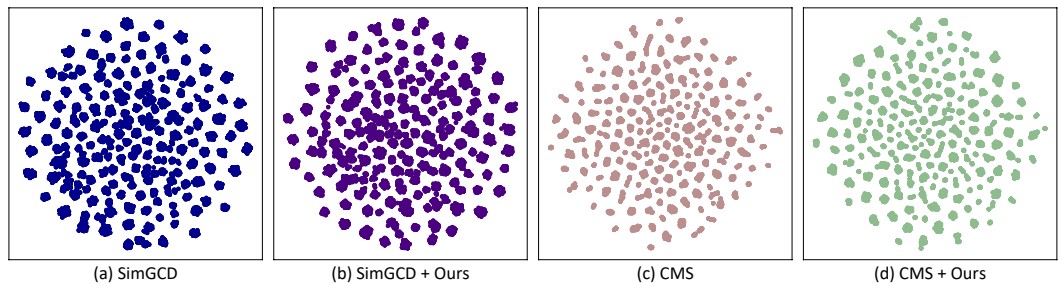

Figure 1: Visualizing the feature distributions of different methods under the CUB dataset.

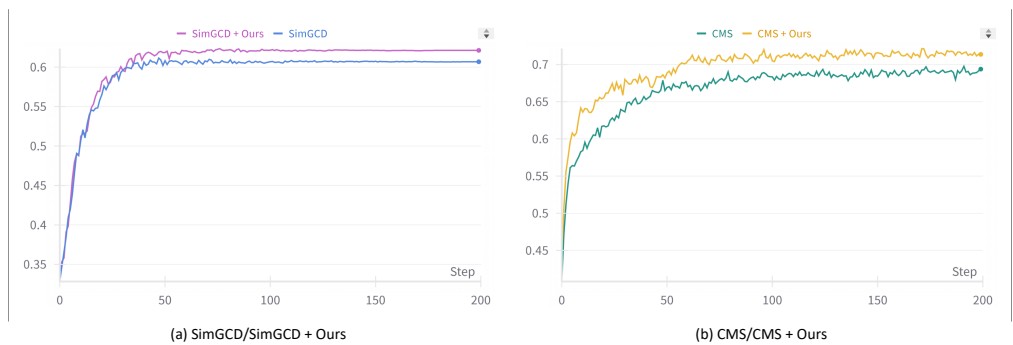

Figure 2: MTMC can improve the convergence speed of the model, which is specifically manifested as a rapid increase in accuracy during the early stages of training.

The implementation of MTMC facilitates the maximization of cls token manifold capacity, enabling high cls token manifold capacity within categories to more accurately occupy positions in feature space during the initial training stages. This effectively reduces confusion arising from the singularity of randomly sampled data during gradient descent and enhances the model's training efficiency. Consequently, employing the MTMC method leads to a substantial improvement in the speed of accuracy enhancement (Figure 2).

# 6 MORE EXPERIMENTAL RESULTS

## 6.1 THE COMPARISON OF ACCURACY

In addition to the above-mentioned experimental content, we also provide experimental results on other different datasets, divided into the fine-grained datasets (Figure 3) and the generic object recognition datasets (Figure 4). It can be found that we perform more consistently on fine-grained datasets, being able to steadily increase accuracy, while there may be a cliff-like decline in accuracy in CI-

FAR100 and ImageNet100. This is because the CMS method may lack discriminability when the dimensional features are too stable.

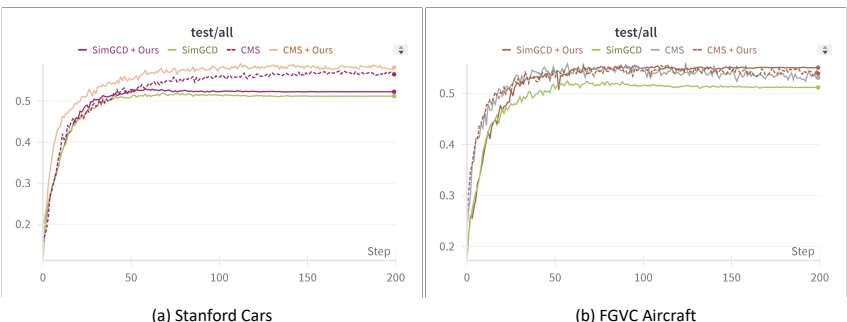

Figure 3: Accuracy of different schemes in training the fine-grained datasets Stanford Cars and FGVC Aircraft.

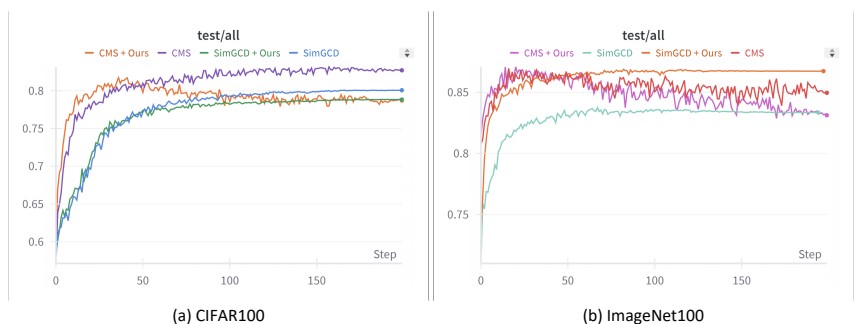

Figure 4: Accuracy of different schemes in training the generic object recognition datasets CIFAR100 and ImageNet100.

## 6.2 THE COMPARISON OF AGGLO

Agglo is an evaluation metric for the degree of feature aggregation. In the baseline method CMS, the weight with the highest agglo score is selected as the best training result. A higher agglo score usually indicates a better evaluation performance, that is, a strong correlation between the inference results and the agglo score. Figures 5 and 6 show the impact of using MTMC on agglo in different characteristic datasets.

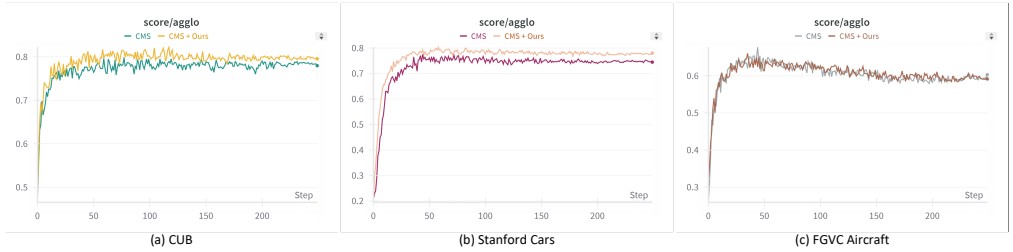

Figure 5: The variation of the metric agglo on the fine-grained datasets CUB, Stanford Cars and FGVC Aircraft.

## 6.3 THE COMPARISON OF ESTIMATED $K$

When inferring without the ground true number of categories, the inference results will be divided into different clusters based on the agglomeration situation, and the predict number of clusters is $K$.

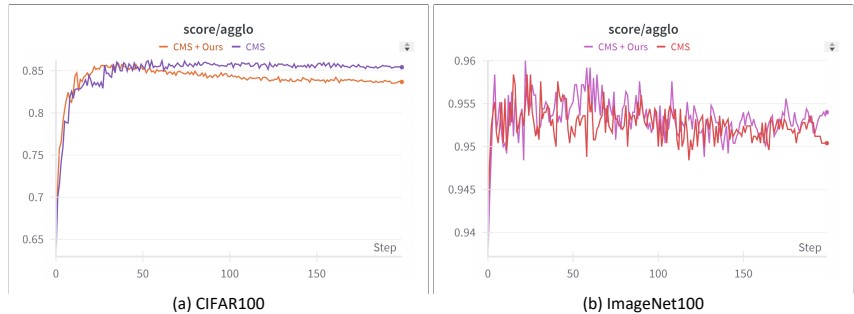

(a) CIFAR100          (b) ImageNet100

Figure 6: The variation of the metric agglo on the generic object recognition datasets CIFAR100 and ImageNet100.

The closer the value of $K$ is to the ground true number of categories, the model does not blindly compress the feature space distribution of each category, making them gather as much as possible. According to Figure 7, it can be observed that using MTMC can improve the evaluation performance of $K$ on most datasets.

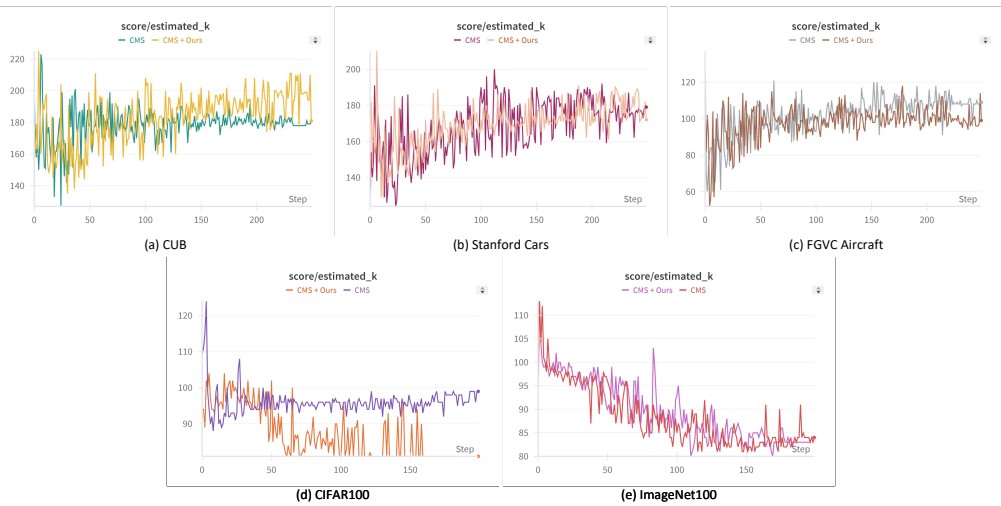

Figure 7: The impact of using MTMC on the predicted number of categories k when the actual number of categories is unknown.