# OpenReview forum: "MTMC: Generalized Category Discovery via Maximum Token Manifold Capacity"
_ICLR.cc/2025/Conference — Submitted to ICLR 2025_

### Official Review · Reviewer_YYUv · 2024-10-20

**Soundness:** 3
**Presentation:** 3
**Contribution:** 2
**Rating:** 5
**Confidence:** 4

**Summary:**

The paper tries to improve the performance for Generalized Category Discovery by leveraging maximum token manifold capacity. The proposed MTMC  employs the nuclear norm to guarantee that the manifolds are both compact and informative. Extensive experiments ensure the effectiveness the proposed method.

**Strengths:**

(1) The paper points out that the the improvement of representation completeness can enhance the performance for GCD.

(2) The proposed method is easy to implement.

(3) The motivation of this paper offers an interesting perspective to solve the GCD tasks.

**Weaknesses:**

(1) It is unclear that why need self-attention to get weighted sample centroid. More experiments need to verify the effectiveness of self-attention in equation 1.

(2) The technical novelty is limited. Maximizing the nuclear norm looks like a general method which can be used in many tasks, such semi-supervised classification. The authors need more explanation about the relationship between GCD and nuclear norm.

(3) In many settings, performance improvements are not significant. And recent methods are not compared , such as [1] and [2].

(4) Limited theoretical analysis about the effectiveness of MTMC mentioned in the first contribution.



[1] SPTNet: An efficient alternative framework for generalized category discovery with spatial prompt tuning

[2] Solving the Catastrophic Forgetting Problem in Generalized Category Discovery

**Questions:**

Please refer to the weaknesses for details.

---

### Official Review · Reviewer_JEXi · 2024-11-02

**Soundness:** 3
**Presentation:** 4
**Contribution:** 4
**Rating:** 6
**Confidence:** 4

**Summary:**

This  paper  introduces  a novel  approach  to  Generalized  Category  Discovery  (GCD) that  emphasizes  maximizing  the  token  manifold  capacity  (MTMC) within  class  tokens.  Unlike  existing  methods  that  focus  on  minimizing  intracluster  variations,  often  at  the  cost  of  manifold  capacity,  MTMC leverages  the  nuclear  norm  of  singular  values  to  preserve  the  diversity  and  complexity  of  data's  intrinsic  structure.  By ensuring  that  different  patches  of  the  same  sample  have  compact  and  low-dimensional  representations,  MTMC enhances  discriminability  and  captures  nuanced  semantic  details.  This  approach  improves  inter-class  separability  and  prevents  the  loss  of  critical  information  during  clustering,  leading  to  a more  comprehensive  and  non-collapsed  representation.

**Strengths:**

1. This paper presents a highly novel and intriguing approach to Generalized Category Discovery (GCD) by leveraging tokens/patches in Vision Transformers (ViT). The introduction of maximizing token manifold capacity (MTMC) offers a significant contribution to the GCD community.

2. The presentation and organization of the paper are excellent. The figures are well-designed and enhance the readability, making the content easy to follow and understand.

3. The experimental results are impressive, demonstrating the effectiveness of the proposed method in enhancing discriminability and preserving the richness of within-class representations.

**Weaknesses:**

1. The authors claim that the tokens of a class can represent intra-class patterns, as shown in Figure 1. However, these tokens are actually features of different patches within an image at the image level. It is not clear how these patch-level features can effectively represent sub-classes at the class level. More detailed analysis or additional experiments are needed to demonstrate this capability.

2. The method for estimating the number of clusters K is not clearly presented. Additionally, in Figure 2, the results are somewhat confusing and do not show an improvement over previous methods in class estimation like GCD/GPC. Do you use the same method with GCD or GPC? Only the best results in Figure 2 should be clearly highlighted in bold.

3. There are some typographical errors in the paper. For example, in line 197, it seems that "equation 4" should be "equation 3."

**Questions:**

See weakness.

---

### Official Review · Reviewer_Uh5s · 2024-11-03

**Soundness:** 2
**Presentation:** 2
**Contribution:** 3
**Rating:** 5
**Confidence:** 4

**Summary:**

The paper presents an approach for Generalized Category Discovery (GCD), by maximizing the token manifold capacity (MTMC) within each class. To be more specific, the MTMC is obtained by employing a nuclear norm to preserve the diversity of the data structure of each class or cluster, thus preventing the representation collapse. Experiments on benchmark datasets show promising performance compared to the listed baseline methods.

**Strengths:**

+ It is interesting to  introduce the maximum manifold capacity on tokens for GCD problem.
+ It is novel to introduce the von Neumann entropy to regualize (or more precisely balance) the [cls].
+ The dimensionality collapse issue is remedy in a degree.
+ The experimental results are promising.

**Weaknesses:**

1.  The presentation is not good. There are a number of confusing or misleading expression / claims.
- While it sounds very interesting to introduce the term [cls], but the reviewer is quite confusing because another concept called the class token manifold extent, or the extend of the sample centroid manifold, is introduced. Is it is essentially the manifold capacity? If yes, why not make the naming consistently?

- In L197-200: There seems a mistake. Eq. (4) cannot be an optimization objective. Should it be Eq. (3)? The reviewer supposed it is the case. However, the interpretation is weird:  "when the sample centroid manifold is maximized, it implicitly minimizes each [vis]
manifold, thereby enhancing the intra-manifold similarity." From the optimization perspective, the reviewer cannot find the equivalence between maximizing Eq. (3) and minimizing each [vis] manifold.

- Similarly, it sounds also misleading in L230: "After training, the manifold capacity increases, compressing the manifold of each sample within the cluster and promoting repulsion between them."


- L393-395: It is vague to connect the "richer and complete representation within each class" and the "distinction bewween different classes". The reviewer cannot see this point. Just a simple instance. In neural collapse, each class is converged to a singleton but different singleton can also be far from each other. From the perspective of Fisher discriminative score, is it having the most discrinimative ability? On the other hand, if the "valume" of each class is expanded, is there a danger to overlapping?

- L420-423: The reviewer was confused what is trying to express.

- L430-L431: The definition of the norm is unclear. It seems not correct to have $\sum_j \lambda_j =1$.

- L463: Is it correct? The reviewer cannot see that claim.

- L466: Does it really a Frobenius norm, or a nuclear norm eventually computed for and illustrated in Figure 4?

- L484-485: The logic behind the sentence is not direct or clear.

2. Though introducing the von Neumann entropy looks intriguing, it suffers from numerical issue because it is more likely some of the eigenvalues are vanishing. Moreover, the definition depends on the rank $k$. However, the rank $k$ is unkown. It is unclear how the performance changes with respect to the parameter of the rank $k$.

3. It is not clear how to estimate the number of categories $K$.

4. The experiments are insufficient. In Figure 3, while it looks flat with respect to the parameter $\lambda$. However, it was a disguise. The range of the parameter $\lambda$ was set extremely tiny, say, 0.001, 0.002, 0.003, 0.004. What about a large range, e.g., [0,1]? This is another implicit parameter $k$ for the "rank". Is it fair to use the groundtruth value or how to estimate the rank? What about the performance with respect to the parameter $\hat k$, which is the estimated numerically?

**Questions:**

1.  Is the concept "extent" essentially the manifold capacity? If yes, why not make the naming consistently?

2. In L197-200: There seems a mistake. Eq. (4) cannot be an optimization objective. Should it be Eq. (3)? How to interprete:  "when the sample centroid manifold is maximized, it implicitly minimizes each [vis] manifold, thereby enhancing the intra-manifold similarity"? From the optimization perspective, the reviewer cannot find the equivalence between maximizing Eq. (3) and minimizing each [vis] manifold.

3. Similarly, it sounds also misleading in L230: "After training, the manifold capacity increases, compressing the manifold of each sample within the cluster and promoting repulsion between them."

4. L393-395: It is vague to connect the "richer and complete representation within each class" and the "distinction bewween different classes". The reviewer cannot see this point. Just a simple instance. In neural collapse, each class is converged to a singleton but different singleton can also be far from each other. From the perspective of Fisher discriminative score, is it having the most discrinimative ability? On the other hand, if the "valume" of each class is expanded, is there a danger to overlapping?

5. L430-L431: The definition of the norm is unclear. It seems not correct to have $\sum_j \lambda_j =1$.  Is it correct?

6. L463: Is it correct? The reviewer cannot see that claim.

7. L466: Does it really a Frobenius norm, or a nuclear norm eventually computed for and illustrated in Figure 4?

9. Though introducing the von Neumann entropy looks intriguing, it suffers from numerical issue because it is more likely some of the eigenvalues are vanishing. Moreover, the definition depends on the rank $k$. However, the rank $k$ is unkown. It is unclear how the performance changes with respect to the parameter of the rank $k$.

10. How to estimate the number of categories $K$?

11. What about the performance with a relatively larger range for $\lambda$, e.g., [0,1]? By the way, where is the $\lambda$ used? The reviewer guess it is to balance the entropy.

12. This is another implicit parameter $k$ for the "rank". How to estimate the rank? What about the performance with respect to the parameter $\hat k$, which is the estimated numerically?

13.  It is not clear how to enable each cluster to prevent dimensionality collapse and enhance the completeness of the representation. Is there a theoeretical justification for this point?

---

### Official Review · Reviewer_ygd5 · 2024-11-03

**Soundness:** 2
**Presentation:** 3
**Contribution:** 2
**Rating:** 5
**Confidence:** 4

**Summary:**

The paper presents a novel approach for GCD that focuses on maximizing the token manifold capacity within class tokens. MTMC aims to enhance intra-class representation completeness by maximizing the nuclear norm of the class token's singular values. The proposed technique emphasizes preserving diversity within intra-class representations and mitigating dimensional collapse, leading to improved clustering performance on known and novel categories.

**Strengths:**

1.The paper provides a well-defined motivation by highlighting how current GCD methods may lead to compressed inter-class distributions and loss of information, impacting clustering accuracy. The concept of maximizing token manifold capacity to improve intra-class representation completeness is innovative and addresses an essential limitation in existing GCD methods.

2.The use of the nuclear norm for enhancing the token manifold capacity is well-founded, with a thorough theoretical explanation supporting its relevance in preventing dimensional collapse.

3.The paper includes experiments on various benchmarks, showing consistent improvements over SOTA methods. The simplicity of incorporating MTMC into existing models is highlighted, which adds practical value for real-world applications.

**Weaknesses:**

1.The introduction of manifold capacity to GCD needs to be explained more specifically. Besides, the method relies on maximizing the nuclear norm to measure the class token manifold capacity. However, maximizing the nuclear norm could introduce computational overhead, especially for large-scale datasets. This raises concerns about the practical efficiency of the approach.

2.While the paper discusses the limitations of embedding quality in datasets like CIFAR100 and Herbarium19, a more comprehensive analysis of scenarios where MTMC may underperform would strengthen the evaluation. While MTMC emphasizes enhancing the manifold capacity, the paper does not provide a detailed comparison with other state-of-the-art manifold learning techniques, such as locally linear embedding or manifold regularization methods. This omission makes it difficult to comprehensively assess its advantages in various contexts.

3.The paper could include more detailed ablation studies to explore how variations in hyperparameters, such as λ (the coefficient for MTMC loss), affect performance across different datasets.

**Questions:**

1.The method involves computing the nuclear norm, which might increase the computational cost, especially for high-dimensional or large-scale datasets. How does the computational efficiency of MTMC compare with existing GCD methods?

2.While MTMC shows effectiveness on visual datasets, how does it perform on non-visual data types, such as text or time-series data? Can the method be easily adapted for these other domains?

3.While maximizing the nuclear norm enhances manifold capacity and representation richness, is there a risk of overfitting? If so, what strategies are proposed to mitigate such risks?

---

### Official Review · Reviewer_vBsT · 2024-11-04

**Soundness:** 3
**Presentation:** 3
**Contribution:** 2
**Rating:** 5
**Confidence:** 4

**Summary:**

This paper introduces a new perspective for the Generalized Category Discovery (GCD) task: Maximum Class Token Manifold Capacity (MTMC). This approach enhances inter-class distinguishability by maximizing the kernel norm of class tokens to expand the diversity and capacity of intra-class features. This method is simple and effective, and can be used as an extensibility component to improve the quality of intra-class feature representation in GCD task.

**Strengths:**

New Perspective: This paper introduces the idea of kernel norm maximization to enhance intra-class feature diversity, thereby improving the performance in the GCD task.

Effectiveness and Extensibility: Experimental results demonstrate that the proposed MTMC component boosts the performance of various GCD methods, showcasing better feature representation capabilities, particularly on fine-grained datasets.

**Weaknesses:**

1. Although this method performs well in terms of performance, its novelty is somewhat limited. The primary innovation lies in applying nuclear norm maximization to the class tokens in ViT for the GCD task. However, the design does not explicitly address the specific needs of the category discovery task but rather resembles a feature expansion enhancement for the general ViT framework.

2. The complexity of Singular Value Decomposition (SVD) is quite high.
Suggestion: If methods such as randomly initialized SVD could be adopted to effectively reduce complexity while maintaining the component’s effectiveness and generalizability, it would further validate the effectiveness of the perspective introduced in this paper.

3. The final loss function appears to focus only on the sum of singular values, which may lead to a concentration of feature expansion in a single direction.
Suggestion: It may be beneficial to consider the distribution of singular values or add some constraints to better align the method with the requirements of the category discovery task.

**Questions:**

1. Could a section comparing computational complexity be added? The complexity of Singular Value Decomposition (SVD) is relatively high.

2. It would be helpful to demonstrate the performance on a dataset with fewer classes, such as CIFAR10.

3. How much improvement does MTMC provide when applied to a more powerful model, like DINO v2?

---

### Meta-Review · Area_Chair_uVwQ · 2024-12-19

**Metareview:**

This paper presents the MTMC to emphasize maximizing the token manifold capacity for generalized category discovery. It can improve inter-class separability without adding excessive complexity. After the response, it receives mixed ratings, including three rejects, and one accept. The advantages, including the clear motivation, use of nuclear norm, and good experimental results, are recognized by the reviewers. However, they are also concerned about the limited novelty, unsatisfying presentation, numerical issues, insufficient experiments, etc. I agree with the reviewers. The current manuscript does not meet the requirements of this top conference. I suggest the authors carefully revise the paper and submit it to another relevant venue.

**Additional Comments On Reviewer Discussion:**

In the discussion, the authors presented detailed responses. However, the main issues are not well addressed. Several reviewers think the proposed response is incomplete and contains many mistakes. All reviewers stick to their original ratings. I think the current manuscript does not meet the requirements of this top conference.

---

### Decision · Program_Chairs · 2025-01-22

Reject